# Postpartum maternal bonding scale: Development and validation in a low- and middle- income country setting

**Bushra Khan** [1‡], **Seyi Soremekun** [2‡], **Waqas Hameed** [3], **Bilal Iqbal Avan** [4]*

1 Department of Psychology, University of Karachi, Karachi, Pakistan, 2 Department of Infection Biology, London School of Hygiene & Tropical Medicine, London, United Kingdom, 3 Department of Community Health Sciences, Aga Khan University, Karachi, Pakistan, 4 Department of Population Health, London School of Hygiene & Tropical Medicine, London, United Kingdom

‡ These authors are joint first authors on this work.
* bilal.avan@lshtm.ac.uk

**Data availability statement:** The data underlying the results presented in the study is available from Dryad: 10.6084/m9.figshare.28377734.

**Funding:** Some of the authors' time, named in the manuscript, is funded by Medical Research Council, United Kingdom. The Reference ID of the grant is: MR/T003375/1 The funders had and will not have a role in study design, data

## Abstract

### Introduction

The World Health Organization's Nurturing Care Framework recommends promoting secure postpartum maternal-infant bonding practices through responsive caregiving for healthy child development. Various instruments exist to assess maternal-infant bonding, but they differ in theoretical foundations and constructs, limiting their broad application and comparability. Notably, there is a lack of bonding instruments developed for low- and middle-income countries (LMICs), where children under five are most at risk of not reaching their developmental potential. This paper describes the development and psychometric validation of a conceptually grounded postpartum maternal bonding scale in an LMIC context and highlights its potential applications in similar settings.

### Methods

Based on a literature review of bonding concepts and measurement processes, we developed a postpartum maternal bonding scale using a cultural adaptation model for psychometric instruments for children and adolescents. This involved identifying and reviewing existing bonding-related tools, generating items, iterative rounds of expert reviews, and pretesting with postpartum women. We then conducted a final survey with a large sample of women at 42 days postpartum to establish the scale's psychometric properties. The study was conducted in the Thatta and Sujawal districts of Sindh, Pakistan.

### Results

An initial pool of 44 items was developed following a literature review and interviews with postpartum women. After multiple rounds of expert review and cognitive pretesting, a 30-item tool was selected for field testing. Using data from 310 postpartum women, we examined the tool's structure through exploratory (EFA) and confirmatory factor analysis (CFA), leading to a refined 12-item tool. The EFA revealed three factors related to

collection and analysis, decision to publish, or preparation of the manuscript.

**Competing interests:** The authors have declared that no competing interests exist.

Emotional, Cognitive, and Behavioural bonding. Taking the four highest loading items from each domain, we performed CFA using three models: a first-order model with the three domains, a second-order model, and a bifactor model, which included an overall bonding construct. The bifactor model showed the best fit (comparative fit index = 0.951; root mean square error of approximation = 0.066; standardized root mean square residual = 0.045). This indicates that both an overall bonding construct and specific domains can be measured separately. Pairwise domain correlations were all below 0.67, and internal reliability statistics ranged from 0.63-0.72 (Cronbach's Alpha) and 0.64-0.77 (global omega). Regression analysis showed associations between bonding scores and factors such as cesarean delivery (reduced behavioural bonding score for mothers having caesarean: -0.94, 95% Confidence Interval -1.86 to -0.01, p-value 0.047), maternal disability (reduced overall bonding score for mothers with severe disability -1.54, 95% CI -3.12 to 0.03, p-value 0.054), and probable postpartum depression (reduced overall bonding score in mothers with probable PPD -1.57, 95% CI -2.70 to -0.45, p-value 0.006).

## Conclusion

The 12-item postpartum maternal bonding scale (PMBS) is a conceptually grounded instrument. It is a brief, easy-to-administer tool with potential cross-cultural use in low- and middle-income settings after cultural adaptation.

## Introduction

Early life experiences from birth to age three have critical significance for the long-term development of human health and well-being. The 2017 Lancet Series on early childhood development estimates the 43% of children under the age of five years in low- and middle-income countries (LMICs) are at high risk of not reaching their developmental potential, and highlights the need for a developmentally appropriate and caring social and physical environment to address this gap [1]. The Nurturing Care framework of the World Health Organisation [2] provides a roadmap for action to promote the holistic development of children, focusing on responsive caregiving, described as the caregiver's ability to understand their child's needs and respond appropriately [3]. Existing evidence suggests responsive caregiving promotes better child health and development [3] and could enhance child-caregiver bonding.

Bonding is a process of forming close relationships between individuals, particularly with reference to the parent and infant it is considered important in developing a parent's unconditional love for the child and the development of a sense of trust in the child. Early bonding in children fosters social competence and optimizes the development of better interpersonal relationships [4–6]. Postpartum maternal bonding, of which responsive caregiving can be a part, is a phenomenon by which a mother develops a strong connection with her infant after birth which generally continues to the first year of the child's life. The postpartum period is the time when a mother, after going through physiologic changes during pregnancy, returns to the non-pregnant state. The phenomenon of bonding involves a range of cognitive, emotional, and behavioral manifestations of the mother that could contribute to the establishment of a secure and nurturing relationship between the mother and infant. Maternal-infant bonding has been linked with improved motor, cognitive, [7,8] social and emotional development [9] in children. However failure to establish a mother-child bond could lead to poor mother-child relationships and child developmental outcomes [10]. There are a number of factors that positively affect postpartum maternal-infant bonding, such as skin-to-skin contact [11], and

rooming-in [12]. However, lack of perceived parental competence [13], lack of social support [14,15] and perinatal poor mental health, particularly depression, may have negative effects on the bonding process [16].

Measurement of the construct of maternal bonding, particularly postpartum bonding, through standardised and valid instruments; is necessary for health service providers, researchers, and policymakers to assess and compare bonding across populations and settings, as well as to evaluate the effectiveness of interventions targeted at enhancing bonding outcomes. We conducted a literature review of existing instruments related to the concept of maternal bonding (S1 Table) and identified key benefits and limitations in their development or use. Of the nine studies identified, most attempted to develop a measure of parent-infant bonding and demonstrate its reliability and/or internal validity, and a few with a basis in a theoretical framework and/or defining of a prior concept [10,17–24]. Only two studies mentioned the use of a theoretical framework to develop their instrument [18,22]. These frameworks were either based on the internal working model of the psychodynamic approach [22] or on parental-fetal emotional attachment during pregnancy [18]. However, these frameworks provide a limited view of the concept of bonding. The internal working model focuses on mental representations a mother has about her child, which can be heavily influenced by her own positive or negative childhood experiences, thus offering a narrow perspective of a mother-child relationship. Similarly, the framework based on parental-fetus attachment may not completely represent postpartum bonding experiences. Additionally, another framework proposed by Goulet [25], which has not been used for instrument development, explains the parent-child interaction as a phenomenon of attachment rather than bonding. It is important to mention that all of these frameworks, whether used in instrument development or not, describe and label the phenomenon as attachment instead of bonding, which leads to confusion regarding the concepts and terminology. Further key issues included are variation in the timeframe for the assessment and focus on only one putative emotional domain of bonding [17,21]; defining bonding in terms of disorder or lack of an explicit definition of the bonding construct or theory [10,20] or definition which did not clearly match accompanying theory [18]; failure to distinguish between pre-and post-natal bonding constructs [21] i.e. same items have been used to assess pre and postnatal bonding. However, in the prenatal phase, the mother would be "longing for" the child rather than bonding with the child, and maternal experiences and behaviours may differ in both phases. Similarly, a focus was given mainly or entirely on the experience of motherhood [19,23] or measuring bonding in adults retrospectively, many years later [20]. Crucially, and importantly for this paper, we could not identify a sufficient evidence base for the development or validation of bonding tools in LMICs [24]. Consequently, the identified heterogeneity in existing instruments could limit their potential for global comparability. These issues are further compounded by a lack of generalisability of current tools to key settings; most of these instruments have been developed in high-income, western contexts, where cultural norms and literacy levels differ significantly from LMIC settings. In LMICs, many mothers may be less educated or more likely to live in rural areas, which could impact their ability to understand and engage with the tools. Additionally, maternal experiences and interactions with children can differ due to factors such as family dynamics and child-rearing-related practices and perceptions, consequently further limiting the applicability of these instruments in the LMIC context

It was noted in the review of the instruments that bonding has been used interchangeably with attachment and also reflected in the assessment of these constructs [18,22,24]. There is evidence that an attachment tool has been used in the past for the assessment of bonding [26]. However, despite confusion between these concepts, recent literature clearly indicates that both are distinct constructs [27,28]. Historically, John Bowlby and Mary Ainsworth pioneered

attachment theory [29,30], which describes how a child naturally seeks emotional relationships with their primary caregiver, driving early relationship formation. Bowlby emphasised that attachment is a child-driven behaviour that seeks security, with a child actively creating internal representations to manage their interactions. Ainsworth extended this by developing and testing four attachment patterns using the 'Strange Situation Test,' emphasising how a child's attachment behaviours influence long-term relationships and have a significant impact on emotional and psychological development throughout their lives. Paediatricians Marshall Klaus and John Kennell, however, first proposed the notion of mother-infant bonding in the 1970s, observing that early interaction between mothers and babies improves parental behaviours and competencies. They suggested the concept of a maternal sensitive phase, which could be the first hours or days postpartum and was considered important in the creation of maternal bonding [31,32]. This idea of a limited period of bonding sparked debate about its scientific validity, and currently, it is considered that bonding can happen throughout the first year of an infant's life [28]. Formally, current literature explains that bonding is a primary caregiver-driven phenomenon which reflects via emotional, cognitive and behavioural ties with the infant [28,33]. It generally appears anytime from pregnancy and continues till one year postpartum. Attachment, on the other hand, is a more specific term. It is a child-driven phenomenon which reflects via emotional, cognitive and behavioural ties with the primary caregiver. It may occur early in life but can be reliably assessed when the child is between 11 and 24 months old [30,34]. The consequences of bonding and attachment differ for both the mother and the child. The establishment of bonding enhances the mother's ability to parent [33], which, in turn, improves the child's survival and development (child consequences) [26]. On the other hand, attachment, which is driven by the child, leads to the child feeling secure in interacting with the caregiver and forming trusting relationships [34]. This, in turn, positively influences the caregiver's communication and relationship with the child (caregiver consequences). Importantly, the measurement of these constructs differs: the caregiver would be the respondent for the bonding [27], while the child would be observed in the measurement of attachment [30]. The interchangeable use of bonding and attachment concepts and terms creates conceptual and linguistic confusion [27], consequently creating challenges in the measurement of these constructs. Therefore, drawing on the existing seminal literature [26–28,30,33–35], we delineated the fundamental differences between the two constructs in Table 1.

Evidence-based models or frameworks for the assessment of postpartum maternal bonding are limited and again tend to emphasize maternal and child attachment [18,25]. Informed by both a thorough review of the literature and a nuanced understanding of the phenomenon of bonding, we have developed a framework that identifies the key psychological constructs relevant to postpartum maternal bonding (please see Fig 1). Through this framework, we defined maternal postpartum bonding as a maternal-driven phenomenon that manifests through emotional, cognitive and behavioural connection between the mother and child in the first year postpartum. The constructs of maternal emotion, cognition and behaviour represent domains of the bonding phenomenon. Each domain has core themes related to maternal bonding. For example, the *emotional domain* focuses on the mother's affective experiences which manifest via her feelings towards the child and her display of emotions during interactions with the child, *cognitive domain* focuses on maternal perceptual and higher mental processes, manifesting through her understanding of and responses to child's needs and cues. The *behavioural domain* focuses on the mother's involvement in the child caregiving process and manifests through her sense of fulfilment and commitment to child care. Each domain has its own psychological expressions and significance and contributes to the overall experience of the mother's bonding with the child.

Table 1. Difference between bonding and attachment with respect to key aspects of these constructs.

| Aspects | Maternal Bonding | Attachment |
|---|---|---|
| **Definition** | **Bonding** is a psychological phenomenon primarily initiated by the caregiver. It involves a mix of emotions, thoughts, and behaviors that reflect the caregiver's developing connection with the child. This connection can begin during pregnancy and often forms quickly after birth through caregiving activities. This bond can be influenced by hormones, pre-birth experiences, and early interactions. | **Attachment**, on the other hand, is driven by the child. It's a complex emotional tie formed over time through the child's interactions and experiences with the caregiver. This attachment reflects the child's feelings of security, trust, and reliance on the caregiver. Unlike bonding, which can happen more rapidly, attachment style and extent develops gradually and is influenced by the caregiver's responsiveness, sensitivity, and consistency in meeting the child's needs. |
| **Timeframe for the manifestation** | Generally appears anytime from pregnancy till one year postpartum | May occur early in life but can be reliably assessed when child is from 11 month to 24 months |
| **Domains with themes** | *Primary caregiver's tie with the child reflected via;*<br>*Emotional*: love; closeness, protectiveness; affectionate acts (display of emotions), pleasure in interaction with the child,<br>*Cognitive*: ability to understand (sensitivity) and responding to child's needs (responsiveness), maternal perception of child behaviour<br>care, monitoring child development<br>*Behavioural*: Concern, sense of fulfillment, commitment to care | *Child's tie with the caregiver reflected via;*<br>*Emotional*: love; proximity seeking, pleasure in interaction with caregiver, anger, fear<br>*Cognitive*: perception of the caregiver, responsiveness to primary caregiver's sensitivity<br>*Behavioural*: approaching or avoiding caregiver |
| **Caregiver consequences** | Improved parenting capacity | Influences nature of communication and connection with the child |
| **Child consequences** | Better survival and holistic development | Ability to form secure and trusting relationships |
| **Current measurement methodology** | Self-rated questionnaires or interview of the primary caregiver | Testing or observation of the child |

LMIC settings have their unique challenges and opportunities that create a need for indigenously developed maternal bonding instrument. Evidence exists that maternal bonding is linked with better child development. In LMIC settings, however, several unique challenges can affect maternal bonding, for example, poor maternal mental health [36,37], poverty, lack of social support, illiteracy and maternal malnutrition [38] may negatively affect a mother's interaction with her child. While, as an opportunity, many early childhood interventions in LMICs tend to emphasise on health, nutrition, education, and sensory stimulation [39,40] but pay less focus on the concept of maternal bonding with the child which could be an important factor in influencing holistic child development.

According to Ettenberger et al. [28] an ideal bonding tool should have a clear definition, focus, timeframe, and methods of assessment. Clarity in assessment is likely to both enhance the validity of the concept and increase its utility in population- based assessment and programming. We, therefore, identify a need for the development of a validated postpartum maternal bonding instrument that addresses evidence gaps - is conceptually grounded, psychometrically sound, suitable for use in LMICs, especially in terms of cultural relevance, cost effectiveness and feasibility and also has potential to be used in various cultures and settings after cultural adaptation.

In this paper, we report on the development of a postpartum maternal bonding scale (PMBS) in a developing country context based on the conceptual framework of psychological constructs of maternal postpartum bonding (Fig 1). We primarily considered recommendations for the measurement of latent constructs [41,42], we focused on five key aspects - (a) the use of a conceptual model (Fig 1); (b) defining criteria for the measurement of the construct; (c) ensuring cultural relevance; (d); statistical validation of the construct (e) and suitable for use in LMICs, considering the context of developing countries.

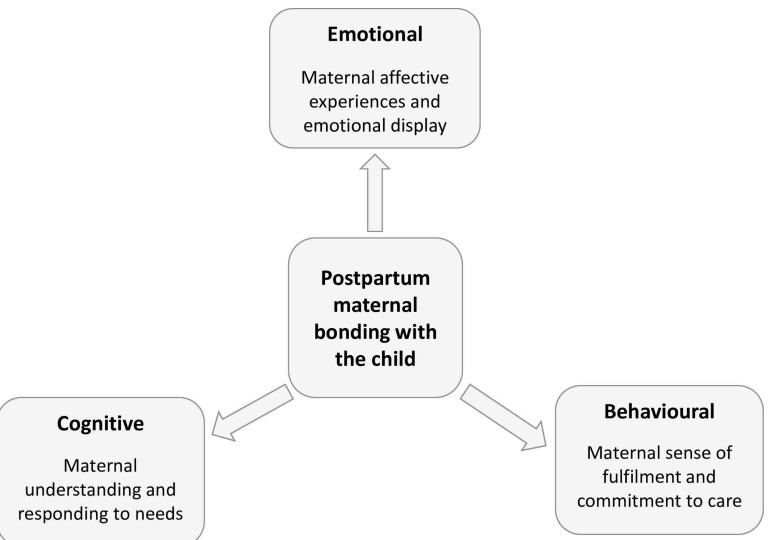

**Fig 1. Conceptual framework of psychological constructs of postpartum maternal bonding.**

## Criteria for measurement

We have added methodological clarity regarding the assessment of the construct of maternal bonding [28] in terms of its definition as emotional, cognitive, and behavioral manifestations of the maternal connection with the infant, its focus (bonding is a maternal-driven phenomenon and it reflects the feelings, thoughts, and behaviours of the mother toward the child); its time frame (postpartum phase: up to 1 year after delivery), and its measurement (based on maternal interview or maternal self-report).

## Cultural relevance

To ensure the cultural and contextual relevance of the instrument we adapted the systematic and iterative model of cultural adaptation for psychometric instruments [43] for the development of postpartum maternal bonding tool. Cultural relevance was particularly ensured through deliberations by the research team and multiple rounds of pretesting the instrument with participants.

Furthermore, to ensure the instrument is conceptually and culturally appropriate, we consulted with relevant experts working in low- and middle- income settings.

## Statistical validation

We used advanced statistical techniques to ensure validation of the construct of postpartum maternal bonding.

## Suitable for use in LMICs

Finally, low-and middle-income countries face professional and financial constraints. Therefore, we ensured that the instrument was feasible in terms of administration, scoring and interpretation of the construct. Cost-effectiveness was also prioritised by developing a brief instrument that could be administered by trained community staff.

To the best of our knowledge, this is the first attempt to develop a clearly defined, conceptually-grounded and culturally relevant postpartum maternal bonding instrument in a developing country setting.

This study aims to describe the process of development and validation of the postpartum maternal bonding scale specifically designed for LMIC settings, addressing the existing gaps in the literature and highlighting its potential application.

## Methods

### Study context

The development and validation of the postpartum maternal bonding scale (PMBS) was part of a large-scale study "Supportive and Respectful Maternity Care' (S-RMC). This study aimed to develop and test the feasibility of a service-delivery intervention model to promote a culture of support and respect during childbirth in public health facilities.

### Scale development

**Study setting and participants.** The study reports on a cross-sectional survey conducted at six secondary-level healthcare facilities in Thatta and Sujawal districts of Southern Sindh, Pakistan. The study was conducted in two phases; scale development and field testing. Forty-seven women were interviewed through convenience sampling in the scale development phase included the stages of items' development, cognitive interviews and pilot testing. Women were recruited at health facilities after childbirth and were later interviewed at their homes at 42 days postpartum. Whereas, to establish psychometric properties during the field testing phase, we carried out a large-scale study titled "Supportive and Respectful Maternity Care' (S-RMC). A total of 310 women were interviewed. Recruitment was limited to women who had given birth at the health facility (study site) during the period of data collection. These women were initially contacted at health facilities and subsequently interviewed in their homes on relevant measures including postpartum maternal bonding, 42 days after delivery. A consecutive sampling technique was used to recruit participants for the study. All women who delivered at the selected facilities during the data collection period were asked to participate in the survey until the desired sample size was achieved. The distribution of the sample across the health facilities was proportional to their childbirth caseloads. Data collection was carried out simultaneously at all six facilities. Only those women who consented to interview at both the health facility and their homes six weeks postpartum were included in the survey. Women living outside the study district or in remote rural areas were excluded due to logistical difficulties and potential security concerns for the data collectors. Women were recruited at health facilities after childbirth and were later interviewed at their homes at 42 days postpartum. All participants provided written informed consent. Only three women were under the age of 18. They were considered emancipated minors because they were married, and therefore parental or guardian consent was not required. The survey was conducted from 19th September 2020 to 14th December 2020.

**Procedure.** The development of the scale followed a systematic and iterative process of item generation, review, and pretesting and piloting before finalisation of the scale for field testing to determine its psychometric properties (Fig 2).

**Items development.** We developed an initial list of items based on; the conceptual framework of psychological constructs of postpartum maternal bonding, assessment criteria for measuring bonding constructs [28], a review of literature and existing instruments relevant to the concept (S1 Table), and interviews with study mothers.

Literature and instruments related to the concept of maternal bonding were reviewed, and relevant items were identified and included in the item pool. In addition to this, five women in the postpartum phase (up to 42 days postpartum) were individually interviewed to gather information about their relationship with their infants, including their thoughts, feelings, and behaviours towards the child pertaining to their emotional experiences with the child, their perception, understanding and fulfilment of the child's needs and finally their commitment to the child care. The information obtained from these interviews was then added to the item pool. Interviews were conducted by a clinical psychologist who had previous experience in conducting interviews.

**Experts' review.**  A pool of items was shared with the core research team, which was composed of multidisciplinary persons with backgrounds in child psychology, maternal and child mental health, psychometrics and implementation science, and the experts for item evaluation along with the conceptual framework and assessment criteria. Three experts had backgrounds in child psychology and psychometrics. Experts and the research team together discussed and reviewed the instrument in terms of construct and content validity, as well as technical and measurement-related aspects of the tool.

**First round of cognitive interviews.**  Cognitive interviews with the women were conducted to explore the comprehension, relevance and cultural appropriateness of the items [43]. Six women at 42 days postpartum were individually interviewed by a trained field staff member with a postgraduate degree in sociology who discussed all items, scoring

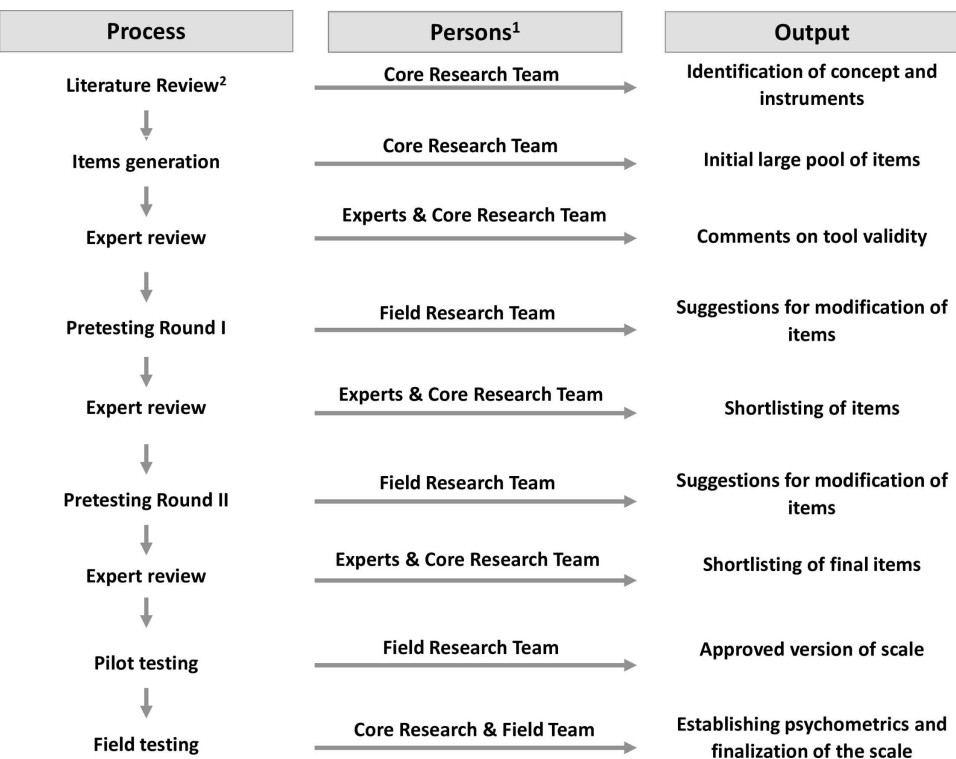

**Fig 2.  Scale development process.** [1]Persons involved in instrument development. The Core Research Team is composed of the developers of the postpartum maternal bonding instrument. The Expert Group was composed of multidisciplinary experts who reviewed the instruments, and finally, the Field Research team were trained interviewers. [2]Findings are summarised in the introduction section. This Figure has been adapted from the Cultural adaptation model for psychometric instruments [43].

mechanisms, and instructions. Study participants' comprehension of items was checked via explanation of items in their own words. Participants were also encouraged to share their views of the meaning, interpretation, relevance and cultural appropriateness of the items. All the suggestions of the participants were noted and shared for the experts' review for incorporation into the draft tool as applicable.

**Second round of experts' review and cognitive interview.** The expert and core teams together carefully reviewed all the suggestions from participants and updated the instrument for the second round of interviews. A modified scale containing 30 items was then pretested via further cognitive interviews with six more women at 42 days postpartum. They were asked to comment on each item in turn with respect to comprehension, relevance and cultural appropriateness.

**Pilot testing of the Postpartum Maternal Bonding Instrument.** All the suggestions given in the second round of interviews were discussed with the experts and core research team members. All suggestions were accepted and the scale was finalised for pilot testing. To explore the ease of administration of the Postpartum Maternal Bonding Instrument, it was then pilot-tested on 30 mothers at 42 days postpartum. After pilot testing and final updates, with the approval of the core team and experts, the instrument and its associated implementation guide were prepared for administration in a large sample of post-partum mothers to explore its psychometric properties.

## Field testing and psychometric assessment

To determine the psychometric properties and construct validity of the 30-item Postpartum Maternal Bonding Scale, a large-scale field test was conducted. In total, 310 women at 42 days postpartum were interviewed at their home using both the draft postpartum maternal bonding tool and a survey questionnaire comprising the below measures. The field team of data collectors was composed of 07 females who had bachelor's degrees and prior experience of working in projects related to maternal and child health and development. The team was supervised by a female field supervisor who had a postgraduate degree in Sociology and an extensive experience in research related to maternal and child health. The team was provided with extensive training on all the instruments based on the instruments' recommended procedures. Data were collected electronically using tablets. The software was developed in Epicollect5 and had a built-in validation check to ensure data quality.

## Measures

### Socio-demographic questionnaire

Socio-demographic questions were related to women/maternal age, education, ethnicity, family type, mode of delivery (if had a cesarean section), child's sex and parity.

### Anxiety and depression

Depression and anxiety were assessed using the Patient Health Questionaire-4 (PHQ-4) developed by Kroenke and colleagues [44]. The PHQ-4 is a brief instrument composed of the PHQ-2 screening tool for depression and generalized anxiety disorders (GAD) screener (GAD-2). PHQ-2 focuses on the two core symptoms of depression according to the Diagnostic and Statistical Manual of Mental Disorders, Fourth Edition (DSM-IV), and GAD-2 collects two core symptoms of anxiety. Participants responded using a 4-point Likert scale; 'not at all' = 0, 'several days but less than one week' = 1, 'more than half the days' = 2, and 'nearly every day' = 3. According to the recommended cut-offs [44–46], positive screening for anxiety was

assigned to participants if the score of the two core symptoms of anxiety was greater than or equal to three (GAD-2 ≥ 3). Similarly, for the two core symptoms of depression, scores greater than or equal to three (PHQ-2 ≥ 3) were assigned positive screening. Lastly, the overall measure of anxiety and depression based on PHQ-4 was categorised with the score as normal (0–2), and as anxiety or depression (3-12).

### Functional disability

Functional disability was assessed via the Washington Groups Disability scale (6 questions) [47]. It has a set of six questions which identify individuals with functional limitations that have the potential to limit independent participation in society. Each question has binary response options -as in 'yes' or 'no'. We have used the overall measure. A response of yes indicated that an individual reported at least some level of difficulty in any of the six domains and would be considered as 'disability or having the risk of severe disability'

### Statistical analysis

Cronbach Alpha and Omega coefficients were used to identify the internal consistency and reliability of the scale. We used Exploratory Factor Analysis and Confirmatory Factor Analysis to identify the factor structure of the scale. We also used a pairwise correlation coefficient to see the correlation between overall bonding and its domains. Finally, we used linear regression to determine the validity of the bonding scale with child and maternal characteristics, including maternal mental health and disability.

### Ethical considerations

Participants' informed consent was obtained after the study's objective, purpose and procedure had been explained to them. The study was approved by the Institutional Review Board of the London School of Hygiene and Tropical Medicine (Reference ID: 17928) and Aga Khan University (Reference ID: 2019-1683-5607).

## Results

### Item development

All the instruments related to the concept of maternal bonding were reviewed and 39 items deemed relevant were included in the item pool. Maternal interviews provided further content for item development for scale domains (Box 1), resulting in the addition of 5 more items in the pool for a total of 44 items.

---

Box 1–Participant responses under key bonding domains in round 1 selected for incorporation into the draft bonding instrument (Item development phase)

**Emotional:** 'I *like to* affectionately touch her...' (Participant P3)

**Cognitive:** *'I play with him…'* (Participant P4)

**Behavioural** *'I try to find information to improve his health and how can I help him grow better and healthy'* (Participant P1). *'Sometimes I feel that I am trapped in this role…'* (Participant P5).

---

### Rounds *of* experts' review and cognitive interview

**Expert review.** Experts, along with the core research team, reviewed the pool of 44 items and endorsed the 4-point Likert scaling of the instrument (categories spanning "never or one-off=0" to "most or all the time=4"). They also endorsed the development and use of the conceptual framework of psychological constructs of postpartum maternal bonding to develop the bonding scale, as it clearly explains the phenomenon of postpartum bonding with the child. During the review, it was decided that i) the tool should be flexible for both self-administration by mothers or interview by a trained staff to account for variation in literacy rates between potential users; ii) the tool was originally developed in English, therefore it was decided that as well as Urdu, the national language of Pakistan, the tool should be translated into Sindhi, the local language of Sindh province, to maximise access and usability. It was a challenge to ensure that all versions of the instruments (in English, Urdu and Sindhi) were equivalent conceptually, semantically and technically. Therefore, the instrument was translated and back-translated by two independent linguists with command over Urdu, Sindhi and English and also adhered to methodological equivalences in the translation process [43]. The original and back-translated versions were then shared with the core research team and the experts for their review. After incorporating their feedback, the instrument was made ready for cognitive interviews.

**Cognitive interviews.** During cognitive interviews, women reported ease with comprehending the tool's instructions and scaling. However highlighted items they felt to be either culturally irrelevant or required modification: For example, the item "*Do you regret having this child?*" received universally negative responses. One participant said '*God gave this child to me….why should I regret this?*'(Participant P5.). Similarly the item, "*Do you feel that you don't have any feelings/emotions towards the child?*" also received uniform negative responses with the consensus this phenomenon simply would not occur: '*I mostly have feelings of love and affection towards the child and very rarely experience feelings of frustration too but never experienced the absence of any feelings*' (Participant P4).

A suggestion to add the term 'vocalize' to the item "*Do you talk with your child*?" was made to accommodate other non-word sounds, as one participant said, '*We talk with the child but sometimes we make only sounds to cater for their attention and to communicate*'(Participant P2).

The core team and experts reviewed all suggestions and agreed to remove items found in interviews to be culturally irrelevant, and update others accordingly for example, "Do you feel that you don't have any feelings/emotions towards your child?" was removed because mothers reported experiencing various emotions toward their child, whether positive or negative. They further decided deletion of items not aligned either conceptually or content-wise with the bonding framework, or which were confusing, ambiguous or duplicated for example, in the cognitive domain, there were two items: (a) "Do you understand your child's signals?" and (b) "Do you understand your child's sleep-wake cycle?" Both items seemed to overlap. During cognitive interviews with mothers, when asked about the first item, they included the child's sleep-wake cycle in their responses and noted that both questions were essentially the same. Therefore, it was decided to keep only the first item, as it already covered the concept, including the sleep-wake cycle as an example. This phase resulted in a 30-item instrument taken to the second round of pretesting. In this round, a small number of semantic-level updates were made to the Urdu and English versions of the tool.

**Pilot testing.** In the pilot phase, both the 30-item Postpartum Maternal Bonding Scale (PMBS) and its implementation guide were tested with field staff and 30 post-partum women. Participants found the instrument simple, culturally relevant and easy to understand, and only minor grammatical changes were made in this phase – similarly, field staff reported ease of use of the guide with no major changes required.

## Field testing

The final 30-item scale used in the field survey of 310 women included 10 items under each domain of Emotional, Cognitive and Behavioural (S2 Table). The characteristics of the expanded sample of 310 women are shown in Table 2. The mean age of the group was 28 years (and 31 years for their husbands). Most women were Sindhi- speaking Muslims (88.1% and 97.4% respectively), with no education (82.6%). Half the sample had a female baby (51.0%) and 11.3% delivered their baby by caesarean. Overall, 12.9% had a risk of severe disability and 31.9% scored over the threshold for at least mild depression. These characteristics are closely resembled with the general population (see S3 Table)

## Psychometric analysis of the postpartum maternal bonding scale after field testing in a survey

**Exploratory factor analysis of the 30-item bonding scale.** An exploratory factor analysis (EFA) of the 30-item PMBS was conducted on the dataset from the final 310 women field survey using Stata V17 (Statacorp, Texas USA). First, items which were highly correlated

**Table 2. Characteristics of the 310 women in the study sample.**

| Characteristic of women | Value |
|---|---|
| **Age in years (mean, SD)** | 28.0 (6.4) |
| **Age category in years (N %)** | |
| 15-24 | 95 (30.7) |
| 25-34 | 149 (48.1) |
| 35-46 | 66 (21.3) |
| **Age in years (mean, SD) - husband** | 31.8 (6.8) |
| **Age category - husband** | |
| 15-24 | 40 (12.9) |
| 25-34 | 144 (46.5) |
| 35-44 | 113 (36.5) |
| 45-54 | 13 (4.2) |
| **Religion category (N %)** | |
| Islam | 302 (97.4) |
| Hinduism | 6 (1.9) |
| Christianity | 2 (0.7) |
| **Mother tongue category (N %)** | |
| Sindhi | 273 (88.1) |
| Urdu, Punjabi, Balochi, Pushto or Sira | 37 (11.9) |
| **Education category (N %)** | |
| None | 256 (82.6) |
| Any formal education | 54 (17.4) |
| **Lives with in-laws (N %)** | 189 (61.0) |
| **Primigravid (N %)** | 42 (13.6) |
| **Last baby delivered by caesarean section (N %)** | 35 (11.3) |
| **Baby was a girl (N %)** | 158 (51.0) |
| **Has a disability (any) (N %)** | 106 (34.2) |
| **Has a disability (severe) (N %)** | 40 (12.9) |
| **Scored in range for mild-severe depression (total 3-12) (N %)** | 99 (31.9) |

were reviewed in a pairwise correlation matrix, and one of the pair with a correlation of 0.8 was dropped. Exploratory factor analysis was performed on the remaining 28 items and, scree plots were generated, suggesting 3 factors (eigenvalues > 1.5) explained 90% of the total variation in the data [48]. Following promax rotation, we kept items which loaded > 0.4 onto any factor. One item which loaded on more than one factor was removed (S2 Table).

Items 1, 4, 5, 6, 9, and 10 loaded onto factor 1 indicating a broad fit to the original compartment of the administered questionnaire relating to the mother's emotional experiences with her child. Item 11 "*Do you feel your child calms down when you pick him/her up?*", originally intended to convey a separate latent construct (cognitive) also loaded the most strongly on factor 1 - suggesting perceived alignment with the *emotional* domain, and likely similarity in interpretation with item 5 "*Do you enjoy holding and picking up your child?*" (S2 Table) which was also most strongly associated with factor 1. Items 13, 16, 19 and 20 were initially developed to identify the strength of the mother's ability to understand and respond to the child's needs. These grouped most strongly onto the second factor indicating consistency in interpretation. Other items (14, 15, 17, and 18) also originally intended for the *Cognitive* domain were less predictive and exhibited relatively stronger loadings on other factors. The final intended domain describes the mother's sense of fulfilment and commitment to child care (*Behavioural*); items 23, 24, 29 and 30 were originally developed to convey this. They particularly interrogated respondents' sense of fulfilment and commitment to caregiving and were appropriately associated with factor 3. Interestingly items 7, 8, 14, and 15 originally responses in *Emotional* and *Cognitive* domains were also associated with this factor. These questions may have been pragmatically interpreted as 'broader feelings and perception about the child' rather than fully conveying the language of the behavioural domain. Other items initially developed under the *Behavioural* domain (items 21, 22, 25, 26 and 28) were more likely to be associated with factor 1; these items explored her sense of commitment to child care. We are cautious to interpret this difference at this stage and note the strength of a mother's commitment to her child may be closely linked with their emotional experience whilst not necessarily measuring the same latent construct.

**Confirmatory Factor Analysis of the 12-item concise bonding scale.** The primary goal in the development of the scale was to produce a reduced-item tool that retained a high predictive value whilst being straightforward to administer and interpretable in low-resource settings. We applied the tool structure identified in the EFA to inform the development of a final 12-item bonding scale by retention of items most strongly associated with the domains, consideration of item redundancy and/or whether items mapped closely to the breadth of the originally intended concepts describing these domains. We kept two items from each domain, which was an a priori decision, to ensure all psychological areas were covered and conceptually balanced. This ensured we achieved a balance between statistically reliable results (statistical fit) and being inclusive of a range of ideas (conceptual breadth). (*Emotional* – emotional experience and emotional display; *Cognitive* –understanding and responding to child's needs; *Behavioural*– a sense of fulfilment and commitment to child care). The resulting scale on which confirmatory factor analysis was performed is shown in Table 3.

Confirmatory factor analysis of the 12-item tool was based on the development of a series of structural equation models in R using the lavaan package comparing three tool structures a) a bonding model consisting of three separate latent 'first order' factors representing *Emotional* (4 items), *Cognitive* (4 items), and *Behavioural* (4 items) which together describe a bonding concept but with no explicit additional 'global' *Bonding* factor, and b) a second order model with an additional global *Bonding* factor predicted by the first order factors; this assumes final scores from all three first-order factors will align in magnitude and direction with the overall bonding score, and c) a bifactor model with four separate first-order latent

**Table 3. 12-item Bonding Scale: factor loadings and goodness of fit statistics from a confirmatory factor analysis (CFA).**

| # | Variable | Factor loadings | | | |
|---|---|---|---|---|---|
| | | First order model | Second order model | Bifactor model (by domain) | Bifactor model (overall bonding) |
| | Factor 1: **Emotional** | | | | |
| 1 | Do you affectionately touch your child? | 0.7738 | 0.801 | 0.716 | 0.480 |
| 2 | Do you enjoy holding and picking up your child? | 0.7770 | 0.804 | 0.650 | 0.570 |
| 3 | Do you feel love for your child? | 0.8010 | 0.802 | 0.669 | 0.621 |
| 4 | Do you feel possessive towards your child? | 0.2626 | 0.282 | -0.046 | 0.593 |
| | Factor 2: **Cognitive** | | | | |
| 5 | Do you feel your child smiles when he/she looks at you? | 0.5710 | 0.601 | 0.627 | 0.142 |
| 6 | Do you vocalize/talk with your child? | 0.7090 | 0.735 | 0.780 | 0.481 |
| 7 | Do you understand your child's signals? | 0.5044 | 0.548 | 0.596 | -0.159 |
| 8 | Do you play with your child? | 0.8470 | 0.850 | 0.883 | 0.309 |
| | Factor 3: **Behavioural** | | | | |
| 9 | Do you feel that taking care of the child is a very difficult task? (Reverse) | 0.3477 | 0.390 | 0.354 | -0.131 |
| 10 | Do you miss the life you had before this child? (Reverse) | 0.2589 | 0.277 | 0.260 | -0.124 |
| 11 | Do you feel trapped after becoming a mother? (Reverse) | 0.7258 | 0.742 | 0.739 | -0.033 |
| 12 | Do you worry that you are not as good as other mothers? (Reverse) | 0.7964 | 0.793 | 0.812 | 0.220 |
| | Overall: **Bonding (Second order only)** | | | | |
| a | Emotional | – | 0.707 | – | – |
| b | Cognitive | – | 0.707 | – | – |
| c | behavioral | – | 0.707 | – | – |
| | **Goodness of fit** | | | | |
| | Comparative Fit Index (CFI) (above 0.90 for CFI) | 0.875 | 0.795 | 0.951 | |
| | Root Mean Square Error of Approximation (RMSEA) (under 0.06 for RMSEA) | 0.089 | 0.111 | 0.066 | |
| | Standardized Root Mean Square Residual (SRMSR) (less than 0.80 for SRMR) | 0.081 | 0.170 | 0.045 | |

constructs of general **Bonding** (all items) **Emotional**, **Cognitive**, and **Behavioural**, which are allowed to vary independently of each other (Fig 3).

Factor loadings for this 12-item tool are shown in Table 3. Likelihood ratio tests suggest all three models are significantly different (first v second order: $\chi^2$ difference = 83.45, p < 0.0001; first v bifactor: 174.26, < 0.0001; second v bifactor: 91.81, < 0.0001). We performed three tests for goodness of fit – the comparative fit index (CFI), root mean square error of approximation (RMSEA) and standardized root mean square residual (SRMSR). Fit statistics and test thresholds are shown in Table 3, indicating the first-order model fit was modest, there is some loss of fit in the restricted (second-order) model and despite some expected loss of strength of factor loadings, good fit across all fitness statistics for the bifactor model.

**Distribution of bonding scores across the sample and inter-domain correlation.** Correlations between the three domains were low (Table 4a), however, the domain scores showed the modest correlation with the overall **Bonding** construct (0.55-0.67). The alpha and omega coefficients of the overall **Bonding** construct and within the domains were in the range of 0.6-0.8 (Table 4b) reflecting modest-to-good internal consistencies [49,50]. Scores for all constructs were created for each woman in the Pakistan dataset from summed (original) Likert values. Score distributions for 310 mothers across bonding domains are presented in Table 4c and Fig 4 indicating participants showed relatively elevated levels of all three bonding domains in relation to their child, particularly for **Emotional** domain.

Double-headed curved arrows: variances of factors or residuals (both heads pointing to same component), or covariances between latent factors (arrow heads pointing to different components). $V_x$: covariance coefficients (omitted in models b and c). MBXX: observed scale items. LX: item loadings on factors from which arrows point. $e_x/i_x$: residual terms.

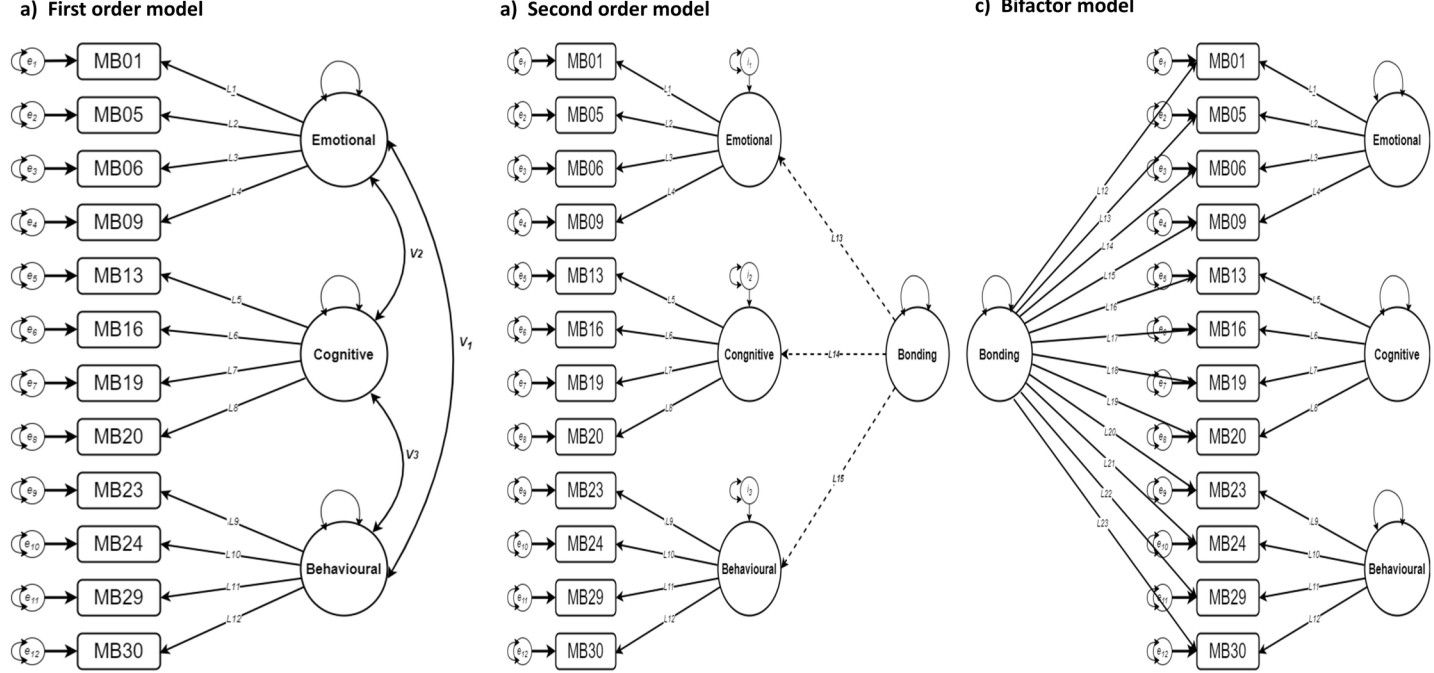

**Fig 3. Confirmatory factor analysis using structured equation models to assess latent constructs in the 12-item Bonding Scale.**

Table 4. **Pairwise correlation coefficients, internal reliability statistics for domains and mean and maximum scores across bonding domains of PMBS.**

| Domain pair | Coefficient | |
|---|---|---|
| **a) Correlation coefficients.** | | |
| Emotional x Cognitive | 0.216 | |
| Emotional x Behavioural | 0.004 | |
| Cognitive x Behavioural | -0.008 | |
| **Bonding** x Emotional | 0.630 | |
| **Bonding** x Cognitive | 0.665 | |
| **Bonding** x Behavioural | 0.554 | |
| **Domain** | **α** | **ω** |
| **b) Internal reliability statistics (α = alpha; ω = omega).** | | |
| Bonding | 0.63 | 0.77 |
| Emotional | 0.74 | 0.77 |
| Cognitive | 0.75 | 0.76 |
| Behavioural | 0.62 | 0.64 |
| **Bonding domain** | **Mean likert score (SD)** | **Max likert score** |
| **C) Mean and maximum scores across bonding domains for 310 mothers.** | | |
| Emotional | 9.76 (2.40) | 12 |
| Cognitive | 7.51 (2.65) | 12 |
| Behavioural | 7.17 (2.63) | 12 |
| α = alpha; ω = omega | | |

**Alignment between maternal and household characteristics and bonding scores.** We used pairwise correlation to assess the correlation between constructs (Table 4a), and linear regression modelsto assess associations with potentially related background and sociodemographic variables. Whilst the three domains of Cognitive, Behavioural and Emotional bonding showed some correlation with the overall Bonding construct (correlation coefficients ranging from 0.55-0.66), lower pairwise coefficients between them suggests three distinct mechanisms contributing to overall bonding that operate relatively independently of each other.

Socio-demographic variables assessed against the Bonding tool were the age of the mother, sex of baby, highest educational level attained, mode of delivery, whether primigravida or not, PHQ-4 depression and anxiety scores and presence of a severe disability.

Linear regression estimates of the relationship between the characteristics of the mother or baby and bonding and its domains are presented in Table 5. If the baby was born by caesarean section, if the baby had severe disabilities, and mothers who had depression/anxiety were all significantly associated with one or more domains in the 12-item bonding tool. Whether this was the mother's first baby showed a trend of association which didn't meet the significance threshold. The findings also indicate that, unlike emotional and behavioural domains, the cognitive domain appeared less affected by the child and maternal characteristics. The cognitive domain focuses on maternal perception and higher mental processes such as problem-solving, sensitivity and responsiveness to the child's needs. Existing literature indicates that maternal sensitivity and responsiveness may not always be impacted by their mental health [51,52], which could explain the lack of significant association observed in our study

## Discussion

The formation of strong maternal-infant bonds is foundational for healthy holistic child development. We developed a postpartum maternal bonding scale (PMBS) to assess the

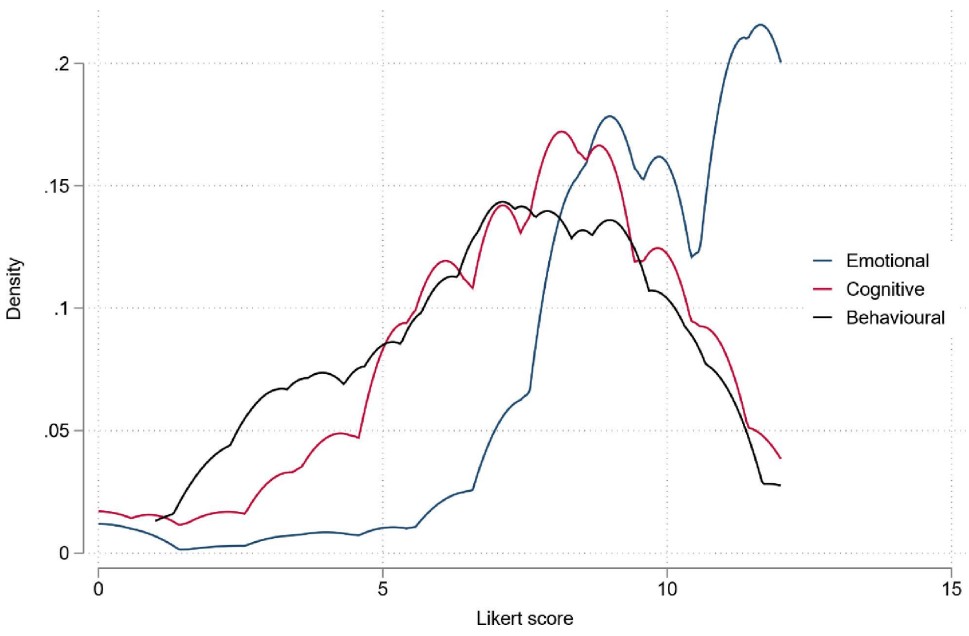

**Fig 4. Density distributions of likert scores across bonding domains for 310 mothers.**

**Table 5.** Univariate regression coefficients and 95% confidence intervals (95% CIs) for the association between summed likert bonding scores and maternal and baby level characteristics.

| Characteristic | Overall: Bonding coefficient and 95% CIs (p-value in brackets) | Domains coefficients and 95% CIs (p-value in brackets) | | |
|---|---|---|---|---|
| | | Emotional | Cognitive | Behavioural |
| **Baby sex** | | | | |
| Boy | 1 | 1 | 1 | 1 |
| Girl | -0.17; -1.23, 0.89 (0.749) | 0.12; -0.42, 0.66 (0.664) | -0.08; -0.67, 0.52 (0.797) | -0.21; -0.81, 0.38 (0.476) |
| **Mother age** | | | | |
| <25 years | 1 | 1 | 1 | 1 |
| 25-34 years | -0.24; -1.47, 0.99 (0.700) | -0.26; -0.88, 0.36 (0.411) | 0.20; -0.49, 0.89 (0.567) | -0.18; -0.86, 0.50 (0.602) |
| 35-46 years | -0.13; -1.63, 1.36 (0.859) | -0.32; -1.08, 0.44 (0.405) | 0.34; -0.49, 1.18 (0.418) | -0.16; -0.99, 0.67 (0.708) |
| Mean age | -0.02; -0.11, 0.06 (0.592) | -0.03; -0.07, 0.01 (0.189) | 0.01; -0.04, 0.06 (0.639) | -0.01; -0.05, 0.04 (0.810) |
| **Mother tongue** | | | | |
| Sindhi | 1 | 1 | 1 | 1 |
| Other | 1.24; -0.38, 2.87 (0.134) | 0.51; -0.31, 1.34 (0.223) | 0.44; -0.48, 1.35 (0.347) | 0.29;-0.61, 1.20) (0.525) |
| **Mother education** | | | | |
| No education | 1 | 1 | 1 | 1 |
| Any formal education | -0.85; -2.25, 0.54 (0.229) | -0.21; -0.92, 0.50 (0.563) | -0.03; -0.81, 0.75 (0.939) | -0.61; -1.39, 0.16 (0.119) |
| **Lives with in-laws** | | | | |
| No | 1 | 1 | 1 | 1 |
| \yes | 0.93; -0.15, 2.01 (0.09) | 0.36; -0.19, 0.91) (0.199) | 0.38; -0.22, 0.99 (0.215) | 0.19; -0.41, 0.79 (0.534) |
| **Caesarean section** | | | | |
| No | 1 | 1 | 1 | 1 |
| Yes | 0.05; -1.63, 1.72 (0.957) | 0.81; -0.03, 1.66 (0.059) | 0.17; -0.77, 1.11) (0.722) | -0.94; -1.86, -0.01 (0.047) |
| **disability (mother)** | | | | |
| None | 1 | 1 | 1 | 1 |
| Risk of severe disability | -1.54; -3.12, 0.03 (0.054) | -1.25; -2.04, -0.46 (0.002) | -0.06; -0.94, 0.82 (0.886) | -0.23; -1.11, 0.65 (0.609) |
| **First child?** | | | | |
| No | 1 | 1 | 1 | 1 |
| Yes | 1.39; -0.15, 2.93 (0.078) | 0.49; -0.29, 1.27 (0.217) | 0.13; -0.74, 1.00 (0.768) | 0.76; -0.09, 1.62 (0.081) |
| **Mother has anxiety or depression** | | | | |
| No | 1 | 1 | 1 | 1 |
| Yes | -1.57; -2.70, -0.45 (0.006) | -0.28; -0.85, 0.30 (0.344) | -0.43; -1.07, 0.20 (0.181) | -0.86; -1.49, -0.24 (0.007) |
| **Anxiety (score ≥ 3)** | | | | |
| No | 1 | | | |
| Yes | 0.43; -0.95, 1.82 (0.541) | 0.44; -0.26, 1.14 (0.217) | 0.20; -0.57, 0.98 (0.609) | -0.21; -0.98, 0.55 (0.589) |
| **Depression (score ≥ 3)** | | | | |
| No | 1 | | | |
| Yes | -2.50; -4.15, -0.85 (0.003) | -0.99; -1.83, -0.15 (0.021) | -0.06; -0.99, 0.88 (0.907) | -1.45; -2.37, -0.54 (0.002) |

Coefficients and p-values (in brackets) from linear regression

phenomenon of postpartum maternal bonding in a developing country context. We developed this scale after an extensive review of the literature and existing relevant tools.

A conceptual framework is very important for the measurement of psychological constructs. Based on the current literature, a framework not only provides structure in identifying its component and their interrelationship but also helps in defining and operationalizing them. There is a scarcity of conceptual models for postpartum maternal bonding that have

the potential to operationalise postpartum bonding. The existing models given by Goulet [25] and Condon [18] have the potential to explain the concept of bonding but they are not distinct and overlap the concept of attachment and bonding, consequently creating confusion/contradiction between their definition and operationalization of the construct. These models need updating in accordance with the current literature related to bonding. Moreover, as a process, it is important to acknowledge that a woman, in postpartum, achieves maternal fulfilment when she bonds with the child. Maternal fulfilment could be explained as a sense of gratification, contentment or finding purpose in her role as a mother. This sense of fulfilment could further enhance maternal connection with the child and could also be an important behavioural indicator of maternal bonding. Based on the contemporary literature we developed a conceptual framework and used and tested it for the development of postpartum maternal bonding in the developing country context.

Our Bonding scale (PMBS) could have relevance to the WHO's concept of responsive caregiving. According to WHO framework for nurturing care, responsive caregiving is the caregiver's ability to notice, understand and timely respond to child's needs [3]; The Cognitive domain in our instrument focuses on mother's sensitivity and responsiveness to child's needs and very much caters to the construct of responsive caregiving, which is primarily defined as a cognitive construct (such as problem-solving ability), We, however suggest the concept may be extended to include emotional (maternal affective experiences with the child) and behavioural (maternal sense of fulfilment and commitment to child care) aspects as well. Our bonding tool encompasses all three elements and could have the potential to be used for the assessment of responsive caregiving in the postpartum phase.

There is, furthermore, a scarcity of postpartum maternal bonding tools developed and validated in LMICs and, to our knowledge, none in Pakistan. PMBS was originally developed in English and back-translated into Urdu and Sindhi. We used the cultural adaptation model [43] as a methodological framework for the iterative tool development process while ensuring multiple equivalences (conceptual, semantic, technical, cultural and content equivalence). Generally, the concept of bonding is assessed either as a positive phenomenon (level of bonding) [18,21] or as a disorder or failure to bond [10,17]. Our scale focuses primarily on the first approach to measure a level of bonding, where we were particularly focused on considering bonding as a positive construct defined as a maternal or primary caregiver-driven phenomenon, which is an affective, cognitive and behavioural manifestation of the primary caregiver's tie with the child. We observed very few bonding-related instruments used psychometric methods that explicitly model the relationships between underlying constructs in instrument development, despite the advantages of such methods and increased access to advanced structural equation modelling methods to do this [42]. Of the studies we identified (S1 Table), few of them used traditional factor analysis/principle components analysis methods to identify constructs [10,18,19,21–23]. We assessed the structure of the tool in our study using exploratory and confirmatory factor analysis techniques, built structured equation models and used post-estimation tests to explore model fit in a dataset of item responses from 310 mothers in the Sindh region of Pakistan. Conceptually, bifactor confirmatory factor analysis - assigning bonding as a global latent variable on which all items load, in addition to loadings of subsets of items on separate domains of emotional, cognitive and behavioural - was the optimal map of the intended tool structure, with relatively good fit compared to the first and second order options. Recommended fit thresholds are under 0.06 for RMSEA [53,54], 0.90 or greater for CFI [54,55] and under 0.80 for SRMR [54,56]. We observed the best fits for SRMR and CFI for the bifactor model despite low factor loadings for items related to cognition, and behavior. Relative loss of fit may be expected in restricted model equations and we interpret these statistical findings cautiously as part of a wider assessment of the needs of the scale and its intended

audience. Nonetheless, our results suggest that the 12-item bonding tool can be used to assess the overall strength of the mother and child bond with additional conceptual factors providing insight into mother's affective experiences (**Emotional**) with her child, her understanding of her child's needs and signals, and her responses to these (**Cognitive**) and her sense of fulfilment and commitment to child care (**Behavioral**).

Our current work highlights the detailed process of tool development and its significant link with maternal mental health and socio-demographic and health factors. Our psychometric assessment indicated an optimal tool structure for our survey sample, but we observed some of the original items did not load on the original domains to which they were assigned, highlighting the difficulty in conveying intended meanings and accurate interpretation by mothers. We therefore retained only items which strongly aligned with the intended constructs from the original development process.

Correlations between domains of the tool were low, highlighting a degree of developmental independence in the activation of the constructs. This was also supported by regression analysis against socio-demographic variables in the dataset - whilst there was no significant relationship between the child's sex, ethnicity, mother education or age of mother and bonding or other domains, we observed significant (p values > 0.05) or close to significant relationships (p values between 0.05 and 0.099) amongst other variables which were frequently domain-specific. This was particularly apparent in the relationship between caesarean section and **Emotional** and **Behavioural**, where mothers who had a caesarean were significantly more likely to have higher mean scores in these domains compared to mothers who had vaginal births, though this indicator was not significantly related to their overall bonding score. The available evidence of the relationship between mode of delivery and mother-child bonding is limited, and results are mixed – a 12-women case study in the United States concluded women who had vaginal deliveries were more likely to strongly recognise the cry of their baby as detected by magnetic resonance images of mothers brains [57] however a larger 2023 study in Germany found stronger mother-child and father-child bonds in caesarean section births [58]. Our study aligns with the second study [58] in the interpretation of the mode of delivery and bonding and addresses a need for equivalent research on the impact of delivery mode on bonding in low-income settings. Maternal/post-partum depression has been previously shown to be associated with bonding difficulties [17,59–61] and we find a similar relationship in our dataset both with overall bonding and domains of **Emotional** and **Behavioural**. It suggests in settings where postpartum screening for depression is applicable, additional screening for the degree of mother-infant bonding with this tool may also be appropriate and valuable. For the first time in an LMIC setting, we observed trends (p values between 0.05 and 0.099) of increased general **Bonding** and women living with their in-laws and primigravid women (also with Behavioural domain) and negative associations between **Bonding** and **Emotional** domain with maternal disability.

In the current study, we interviewed mothers as a primary caregivers however, further research could assess the validity of the tool when applied to other caregivers of the child. We also assessed bonding at 42 days postpartum (6 weeks) with the rationale that this is the average period by which women, after a normal delivery, recover from reproductive changes experienced during pregnancy, allowing for the potential phenomenon of bonding to be matured enough to be assessed or inquired about. We developed the scale to be applicable for up to one year postpartum. Although the current study collected data from women at 42 days postpartum, it is recommended that this scale be tested at other time points post-delivery, considering that recovery during postpartum time can sometimes last up to 6 months [62] and its psychological effects can continue for one year, such as postpartum depression [63]. Therefore, as bonding is a psychological construct, its assessment may be conducted up to one

year postpartum. However, the WHO clearly emphasises the significance of perinatal mental health in maternal and child-related initiatives [64]. Evidence exists showing its link with the mood, which could influence the mother's bond with the child [65–67] and her caregiving capacity. Therefore, assessing bonding in the early postpartum period might be more useful in determining the bonding level and providing timely interventions to improve the maternal-child bond. This, in turn, could contribute to improved child growth and developmental outcomes.

For service provision or population-level studies of bonding and related phenomena, brief and easy-to-administer assessment scales are desirable. Using a mix of field and statistical methods we reduced an initial pool of 44 items to our final brief scale with the intention for its use in research and related contexts, including health services and programmes related to postpartum bonding, responsive parenting and early childhood development monitoring. Successful administration of any psychometric instrument is however dependent upon a clear and standardized protocol to guide the user in its administration and the interpretation of findings. We developed an implementation guide for the Postpartum Maternal Bonding Scale (PMBS) which includes the definition of the concepts, purpose and description of the instrument. It further provides guidance with respect to types of respondents, format, administration, scoring and the interpretation of the scores. We have provided the guide and the final versions of the instrument in English and Urdu (Please see S1 File for the implementation guide and instrument). In addition to this, in our scale, an increasing score in overall bonding and its domains indicates better maternal postpartum bonding with the infant. However, it is important to highlight that attaining a score of zero on any domain warrants particular concern as it signifies a complete absence of that aspect of bonding. Therefore, attention should be given to such cases, along with those who have low scores – both overall and in specific domains.

We finally identify several possibilities for extended study of this scale. Further construct validity of the bonding instrument may be established by conducting confirmatory analyses in other samples as we performed both exploratory and confirmatory factor analyses in the same group of women. Moreover, we had a reasonable and sufficient sample size from a psychometric scale development perspective. However, larger sample size studies would be needed to establish the feasibility, implementation and use of this scale in epidemiological and public health programmes. Other validity tests may also assess its association with child growth and development outcomes and determine acceptable cut-offs. Scale may be used with similar populations and settings. The study can be replicated in other low-and middle-income countries after cultural adaptation of the scale.

## Conclusion

We have developed a 12-item postpartum maternal bonding scale (PMBS) based on conceptual foundation to create a theoretically valid instrument that has been developed in the context of low-and middle-income countries. It is a brief, easy-to-administer scale with the potential for cross-cultural use after adaptation.

## Supporting information

**S1 Table. Table of review of maternal bonding related instrument.**
(PDF)

**S2 Table. List of Items in the 30-item Maternal Bonding Scale.**
(PDF)

S3 Table. **Comparison between study sample and general population.**
(PDF)

S1 File. **Implementation guide and English and Urdu version of Postpartum Maternal Bonding Scale (PMBS).**
(PDF)

AcknowledgmentWe are grateful to the experts for their input. We are thankful to Faiza for supporting us in pretesting and Ghani in the field testing of the scale. We are also thankful to Rabeea and Hiba for helping us in the initial literature search. Most importantly, we are immensely grateful to the mothers who participated in this study.

## Author contributions

**Conceptualization:** Bushra Khan, Seyi Soremekun, Bilal Iqbal Avan.

**Data curation:** Waqas Hameed.

**Formal analysis:** Bushra Khan, Seyi Soremekun.

**Funding acquisition:** Waqas Hameed, Bilal Iqbal Avan.

**Investigation:** Bilal Iqbal Avan.

**Methodology:** Bushra Khan, Bilal Iqbal Avan.

**Project administration:** Bushra Khan, Waqas Hameed.

**Software:** Seyi Soremekun.

**Supervision:** Bushra Khan, Waqas Hameed, Bilal Iqbal Avan.

**Writing – original draft:** Bushra Khan, Seyi Soremekun.

**Writing – review & editing:** Bushra Khan, Seyi Soremekun, Waqas Hameed, Bilal Iqbal Avan.

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
