## [Decision Letter · Decision Letter 0]

28 Aug 2024

PONE-D-24-27766
Postpartum Maternal Bonding Scale: development and validation in a low- and middle- income country setting
PLOS ONE

Dear Dr. Hameed,

Thank you for submitting your manuscript to PLOS ONE. After careful consideration, we feel that it has merit but does not fully meet PLOS ONE’s publication criteria as it currently stands. Therefore, we invite you to submit a revised version of the manuscript that addresses the points raised during the review process.
 
Thank you for submitting your paper to our journal. We appreciate your efforts in ensuring that assessment measures are effective in low- and middle-income countries (LMICs). Your work is commendable.

The two reviewers have provided valuable feedback. Please address their comments and suggestions in your revised manuscript.

We look forward to receiving your revised manuscript.

Kind regards,

Maiken Pontoppidan

Academic Editor

PLOS ONE

Reviewers' comments:

Reviewer's Responses to Questions

**Comments to the Author**

1. Is the manuscript technically sound, and do the data support the conclusions?

Reviewer #1: No

Reviewer #2: Partly

2. Has the statistical analysis been performed appropriately and rigorously? 

Reviewer #1: Yes

Reviewer #2: Yes

3. Have the authors made all data underlying the findings in their manuscript fully available?

Reviewer #1: Yes

Reviewer #2: Yes

4. Is the manuscript presented in an intelligible fashion and written in standard English?

Reviewer #1: Yes

Reviewer #2: Yes

5. Review Comments to the Author

Reviewer #1: Dear Editor thank you for your invitation to review manuscript entitled" Postpartum Maternal Bonding Scale: development and validation in a low- and middle- income country setting"

Overall, the introduction provides a solid foundation for the research.

It clearly outlines the significance of early maternal bonding, highlights the research gap in LMIC settings, and states the study objectives. However, there are a few areas where it could be strengthened:

Cultural Sensitivity:

1. While the introduction mentions ensuring cultural relevance, it might be beneficial to elaborate on specific strategies used to adapt the scale to different cultural contexts.

Consider consulting with experts from the target LMICs to ensure the scale is culturally appropriate and sensitive.

2. Strengthen the Thesis Statement:

Consider refining the thesis statement to more directly state the purpose of the study. For example, you could state: "This study aims to develop and validate a postpartum maternal bonding scale specifically designed for LMIC settings, addressing the existing gaps in the literature."

3. Expand on the Literature Review:

While the literature review provides a good overview of existing instruments, it could be further enhanced by:

Discussing the theoretical frameworks underpinning these instruments.

Comparing and contrasting the strengths and limitations of different scales.

Analyzing why these existing scales might not be fully applicable in LMIC settings.

4. Clarify the Conceptual Framework:

Explicitly define the concept of postpartum maternal bonding, including its key components and dimensions.

Discuss how this definition aligns with existing theoretical frameworks (e.g., attachment theory).

5. Strengthen the Discussion of LMIC Context:

Elaborate on the unique challenges and opportunities for maternal bonding in LMIC settings.

Discuss how cultural, social, and economic factors might influence the development and validation of the scale.

Method

1. Sample Size and Representation:

• Discuss the representativeness of the sample in terms of age, education level, socioeconomic status, and other relevant demographic factors.

• Consider providing more information on the recruitment process and inclusion/exclusion criteria.

2. Cultural Adaptation Process:

• Elaborate on the specific strategies used to adapt the scale to the LMIC context, such as back-translation, cognitive interviews, and expert review.

• Discuss any challenges encountered during the cultural adaptation process.

3. Psychometric Properties:

• Provide more details on the statistical techniques used for construct validity, reliability, and internal consistency analysis.

• Report the Cronbach's alpha coefficient and factor analysis results.

Results:

1. Factor Analysis and Model Fit:

Provide more details on the criteria used to determine the number of factors to extract in the exploratory factor analysis (EFA).

Discuss the implications of the bifactor model fit in terms of the scale's underlying structure and interpretation.

Report the full details of the confirmatory factor analysis (CFA) models, including the specific goodness-of-fit indices and their cut-offs.

2. Cultural Adaptation and Translation:

Elaborate on the specific strategies used to ensure the equivalence of the scale across languages (e.g., back-translation, expert review).

Discuss any challenges encountered during the cultural adaptation process and how they were addressed.

3. External Validity:

Consider discussing the potential for external validity by comparing the scale's findings to those from other studies or existing instruments.

Explore the generalizability of the scale to different populations and settings.

Reviewer #2: Thank you for the opportunity to review the manuscript “Postpartum Maternal Bonding Scale: development and validation in a low- and middle-income country setting”. I read it with great interest, and I applaud the huge work the authors have done in developing and validating an instrument. Especially in a setting that hasn’t been studied much. However, I also have some comments for the authors:

Abstract

1. Ll. 55 “Regression analyses showed associations…”: The authors should state the direction of the associations.

2. L. 61: I would be cautious not to overstate the instrument’s potential cross-cultural use, as the authors have only tested it in one region.

Introduction

3. Ll. 118-119: I do not agree with the statement that attachment appears at 11-24 months. Rather, it is when it can be reliably assessed with the Strange Situation Procedure.

4. Table 1: I am unclear on what the authors mean with a lot of these differences between attachment and bonding, like caregiver consequences and child consequences. I do not see why those examples are particular to bonding and attachment. References are sorely needed.

5. I am not sure I understand the difference between the emotional and behavioral domains in the authors’ figure. As an example of behavioral, the authors mention the mother’s sense of fulfilment, however, isn’t that also an emotion and could be in the emotional domain? The authors mention that this is based on a thorough review of the literature, but it would be helpful if the authors could review this literature more thoroughly in the manuscript as well.

Results

6. Ll. 317: Could the authors give some examples of the items that were removed to create transparency in the process?

7. Ll. 386: What thresholds did the authors use to evaluate the goodness of fit estimates? I see some are mentioned in Table 2, but references on them would be good.

8. There are two Table 3s: One about correlations and reliability and one about descriptive statistics.

9. Regarding the second Table 3 on the descriptive statistics and Figure 4 on the density distribution, I would be interested in knowing the distribution of scores on all the individual items. Do the authors see low scores on any of the items? Because if not, it may be an indication of the sample not being representative in giving low scores, challenging the validation of the instrument.

10. I would urge the authors to consider using item response theory to also choose which items to retain or not rather than just relying on expert/participant evaluations and loadings in the exploratory factor analysis. The strength of using item response theory is that it can be used to evaluate at which levels of the construct that the item discriminates. That way the authors can be sure that they have items that assess the construct at all levels of the trait.

11. P. 19: The authors mention that primigravida showed a trend towards significant. However, there are many other instances of the p-value being above the significance level, so I am unsure why the authors single out this one.

12. Table 4: I would encourage the authors to also report – at minimum – the confidence levels.

13. Table 4: Why have the authors categorized mother’s age into three categories? Continuous variables retain more power, so is there a specific reason why it is handled as a categorical variable instead?

14. P. 19 “We used pairwise correlation to assess the correlation between constructs…”: Do the authors mean the subdomains? And I cannot find any report of these analyses in the manuscript.

Discussion

15. P. 27: The authors mention existing models by Goulet and Corden, but they haven’t been introduced earlier in the background. This is also related to my earlier comment that I am missing a review of the literature on maternal bonding from the authors.

16. As a limitation, I think the authors should address the fact that both the exploratory and confirmatory factor analyses were done in the same sample.

17. P. 29 “It may also be helpful to further consider the value of the first and second order model options”: Interesting, could the authors expand on this point?

18. P. 29-30: Could the authors give some interpretations on why they don’t find any significant associations with the cognitive subdomain?

19. P. 30: I find it unclear whether the authors think that their instrument is valid to assess bonding up to one year postpartum. As the authors have only tested the instrument at one time point, I would be cautious not to overstate the use of the instrument.

Other comments

20. The manuscript would benefit from additional proof-reading. There are several overly long sentences and verb tenses in the same sentence that makes reading challenging.

6. PLOS authors have the option to publish the peer review history of their article (what does this mean?). If published, this will include your full peer review and any attached files.

Reviewer #1: **Yes: **no

Reviewer #2: No

---

## [Author Response · Author response to Decision Letter 1]

24 Oct 2024

A separate file is uploaded that provides point-by-point response to the reviewers comments

---

## [Decision Letter · Decision Letter 1]

27 Nov 2024

PONE-D-24-27766R1
Postpartum Maternal Bonding Scale: development and validation in a low- and middle- income country setting
PLOS ONE

Dear Dr. Hameed,

Thank you for submitting your manuscript to PLOS ONE. After careful consideration, we feel that it has merit but does not fully meet PLOS ONE’s publication criteria as it currently stands. Therefore, we invite you to submit a revised version of the manuscript that addresses the points raised during the review process.

Thank you for resubmitting your manuscript, “Postpartum Maternal Bonding Scale: Development and Validation in a Low- and Middle-Income Country Setting,” for further consideration. We appreciate the revisions and effort you have put into addressing the reviewers’ comments.

Upon reviewing the revised manuscript, we note that while substantial improvements have been made, one of the reviewers has suggested that the manuscript would benefit from a more thorough proofread to address language issues and improve overall clarity.

We encourage you to carefully review the text. You may also consider seeking professional language editing services to ensure that the manuscript is polished and ready for publication.

Once these revisions are complete, we will be happy to proceed with the final evaluation.

Thank you for your continued efforts and contributions to the field. Please do not hesitate to reach out if you have any questions or require clarification regarding the next steps.

We look forward to receiving your revised manuscript.

Kind regards,

Maiken Pontoppidan

Academic Editor

PLOS ONE

Journal Requirements:

Reviewers' comments:

Reviewer's Responses to Questions

**Comments to the Author**

1. If the authors have adequately addressed your comments raised in a previous round of review and you feel that this manuscript is now acceptable for publication, you may indicate that here to bypass the “Comments to the Author” section, enter your conflict of interest statement in the “Confidential to Editor” section, and submit your "Accept" recommendation.

Reviewer #1: All comments have been addressed

Reviewer #2: All comments have been addressed

2. Is the manuscript technically sound, and do the data support the conclusions?

Reviewer #1: No

Reviewer #2: Yes

3. Has the statistical analysis been performed appropriately and rigorously? 

Reviewer #1: Yes

Reviewer #2: Yes

4. Have the authors made all data underlying the findings in their manuscript fully available?

Reviewer #1: Yes

Reviewer #2: No

5. Is the manuscript presented in an intelligible fashion and written in standard English?

Reviewer #1: Yes

Reviewer #2: Yes

6. Review Comments to the Author

Reviewer #1: Accepted. No need more revision. All comments have been addressed. Manuscript can be published in PLOS ONE.

Reviewer #2: Thank you to the authors for their thorough responses to my comments. I believe that the paper has been strengthened considerably.

My one remaining note is that although the authors have proof-read the manuscript, I would recommend them to give it one more glance, as there are still some grammatically issues and overly long sentences. For example:

Introduction, P. 5, l. 104: “Of the nine studies identified” instead of “Of nine studies identified”

Introduction, P. 7, l. 155: “It generally may occur” instead of “It generally may occurs”

Results, P. 23: Ll. 505-513 is all one sentence and too long to be easily read and understood.

Results, P. 24, l. 526: It should be “Correlations” not “Correlation”

Results, P. 24, l. 535: “linear regression models” instead of “regression methods”

Discussion, P. 38, L. 712: “…phenomenon of bonding to be matured” instead of “…phenomenon of bonding to matured”

Discussion, P. 39, Ll. 737: “…attaining a score of zero…” instead of “…attaining zero score…”

Discussion, P. 40, Ll. 750: Again, I think that the authors should be more cautious in their language. The study may be replicated in other settings, but it is not certain that it can – we would need to conduct these replication studies before we are sure that the study can be replicated.

7. PLOS authors have the option to publish the peer review history of their article (what does this mean?). If published, this will include your full peer review and any attached files.

Reviewer #1: No

Reviewer #2: No

---

## [Author Response · Author response to Decision Letter 2]

23 Dec 2024

A separate file is uploaded that provides point-by-point response to the reviewers comments

---

## [Editor Report · Decision Letter 2]

8 Jan 2025

Postpartum Maternal Bonding Scale: development and validation in a low- and middle- income country setting

PONE-D-24-27766R2

Dear Dr. Hameed,

We’re pleased to inform you that your manuscript has been judged scientifically suitable for publication and will be formally accepted for publication once it meets all outstanding technical requirements.

Kind regards,

Maiken Pontoppidan

Academic Editor

PLOS ONE

Additional Editor Comments (optional):

Thank you for resubmitting your manuscript titled "Postpartum Maternal Bonding Scale: Development and Validation in a Low- and Middle-Income Country Setting".

We appreciate the effort you have taken to address the reviewers' comments and make revisions to your work.

The revised manuscript will now proceed to the next stage of the review process.

Thank you again for your contribution to the journal, and we look forward to finalizing this work for publication.
---

## [Editor Report · Acceptance letter]

PONE-D-24-27766R2

PLOS ONE

Dear Dr. Hameed,

I'm pleased to inform you that your manuscript has been deemed suitable for publication in PLOS ONE. Congratulations! Your manuscript is now being handed over to our production team.

Kind regards,

on behalf of

Dr. Maiken Pontoppidan

Academic Editor

PLOS ONE